# Geographic Variability of Sugars and Organic Acids in Selected Wild Fruit Species

**DOI:** 10.3390/foods9040462

**Published:** 2020-04-08

**Authors:** Asima Akagić, Amila Vranac Oras, Sanja Oručević Žuljević, Nermina Spaho, Pakeza Drkenda, Amna Bijedić, Senad Memić, Metka Hudina

**Affiliations:** 1Faculty of Agriculture and Food Sciences, University of Sarajevo, Zmaja od Bosne 8, 71 000 Sarajevo, Bosnia and Herzegovina; vranac.amila@gmail.com (A.V.O.); s.orucevic-zuljevic@ppf.unsa.ba (S.O.Ž.); n.spaho@ppf.unsa.ba (N.S.); p.drkenda@ppf.unsa.ba (P.D.); bijedic.amna@gmail.com (A.B.); sbhsarajevo@gmail.com (S.M.); 2Department of Agronomy, University of Ljubljana, Biotechnical Faculty, Jamnikarjeva 101, 1000 Ljubljana, Slovenia; metka.hudina@bf.uni-lj.si

**Keywords:** wild fruit, sugars and organic acids, HPLC analyses, processing, pedoclimatic conditions

## Abstract

The chemical variability of bilberry (*Vaccinium myrtillus* L.), wild strawberry (*Fragaria vesca* L.), cornelian cherry (*Cornus mas* L.) and rosehip (*Rosa canina* L.) based on the content of individual and total sugars and organic acids in fruit was investigated. The fruits were picked in fully ripened condition within the period from 2014 to 2015 from different locations. The fresh fruits were analyzed with the high performance liquid chromatography (HPLC) technique for the purpose of identifying and quantifying the content of glucose, fructose and sucrose, as well as malic, citric, fumaric and shikimic acids. However, the content of individual sugars and organic acids differed by locations as well as by growing year within the same wild fruit species. The differences between wild fruit species as well as among different locations are presented by principal component analysis (PCA). Based on results obtained, rosehip fruits with higher sugars and organic acids ratio (S/A) are suitable for production of “pekmez” and drying, while genotypes of cornelian cherry, wild strawberry and bilberry with lower S/A are recommended for production of juices and gelatin products. The research results show that specific environmental conditions may influence significantly the content of analyzed parameters, as is the case with cornelian cherry and rosehip. Considering that the food industry is searching for new products, the wild fruit species analyzed represent a promising source of ingredients for the development of beverages and foods with functional properties as well as for supplements and nutraceuticals.

## 1. Introduction

Traditional wild fruit is an important foodstuff in the local population′s nutrition, and it has been used in folk medicine since ancient times. Increased interest in wild fruit results primarily from the need for biodiversity preservation and use as a potential source of new nutraceuticals aimed at prevention of a series of diseases [1,2]. In addition, it improves the living conditions of the population in local rural areas, who exclusively engage in collecting and selling wild fruit and medicinal herbs, which are often their only sources of income [3]. The nutritionally valuable wild fruit, exhibiting high tolerance to different ecological and pedological conditions, and diseases and pests [4,5], is highly appreciated by consumers. This fruit, in comparison with cultivated ones, contains a higher concentration of bioactive components [6,7,8,9,10], which have a positive impact on health [11,12,13,14,15]. However, its chemical composition varies, and it is under the influence of a series of factors before and after harvesting [16,17,18,19,20,21,22,23]. Research of climate differences conditioned by geographical environment is very important from the perspective of comprehending the differences in fruit quality (content of sugars, organic acids, bioactive components etc.) and for optimizing the ripening stage [24,25,26,27,28], which are essential parameters of fruit quality.

Sugars and organic acids are the most common soluble constituents of fruit. They have an important influence on taste, shelf life and nutritive properties, and they are reliable indicators of acceptability by consumers [29,30,31,32]. Changes in sugar and organic acid composition and concentration are reflected also on changes of fresh fruit quality and their products [29]. From a technological perspective, they are very important for formation of gel consistency in gelatin products [33] and in production of juices and nectars, since they define sweetness index [34]. The representation of individual sugars and organic acids serves also as an indicator of authenticity of fruit products [35,36,37]. In addition, sugars participate in polyphenol biosynthesis [38]; thus, higher sugar content in fruit implies higher polyphenol concentration [39,40], which is exceptionally important from the perspective of a nutrient-enriched diet. 

As far as biodiversity of forest flora is concerned, Bosnia and Herzegovina (B&H) is among the richest countries in Europe [41]. Most of the forest plant species identified in B&H, according to World Health Organization (WHO) recommendations, may satisfy daily needs for basic nutrients, especially vitamins A and C, and some mineral substances [3]. According to research of Marjanovic-Balaban et al. [42], wild berries originating from B&H, due to appropriate climate conditions and altitude, are very rich in chemical components, which have a protective impact on consumer’s health. Based on the fact that wild fruit species such as rosehip, cornelian cherry, wild strawberry and bilberry occur widely and are economically the most important, having a great market value in B&H, they were selected for investigation in this paper. Furthermore, the fruits of wild strawberry and bilberry are processed by the local population into valuable end products such as juice, jam and marmalade, and reach high prices on the market, which also improves their living conditions. According to recent literature, there are no data on sugar and organic acid profiles of wild strawberry and bilberry fruits collected at altitudes above 1000 m. Much of the literature focuses on the analysis of basic parameters of chemical composition of this valuable fruit in B&H [42,43,44,45,46]. However, there are no studies where the content of individual sugars and organic acids is determined in relation to wild fruit, except for cornelian cherry [19,47], and these are important parameters of quality of fruit and fruit products. Therefore, this paper aims at (i) determining individual sugars and organic acids content in wild fruit species, (ii) establishing the influence of location and growing year on fruit quality parameters, (iii) recommending to producers the processing of wild fruit species according to the sugars and organic acids ratio (S/A) and (iv) analyzing the relationship of individual locations’ pedoclimatic data and chemical parameters of wild fruit species.

## 2. Materials and Methods

### 2.1. Fruit Material

Meteorological data were obtained from different locations in Bosnia and Herzegovina where fruit from naturally occurring flora (rosehip (dog rose), cornelian cherry, wild strawberry and bilberry) without anthropogenic influence was picked during 2014 and 2015 (Table 1). These wild fruit species are widely distributed and economically the most important, having great market value in B&H. 

Selected locations are the best known and are very important for wild fruit production in B&H. The fruit was picked in the optimal stage of ripeness for processing (from the end of June to the end of October), which was established based on color (Table 2). Immediately after harvesting, the samples were transported to the laboratory where only undamaged fruit was selected from each location (five independent samples every year), 1 kg for each wild fruit species. All samples were immediately frozen in liquid nitrogen and kept at a temperature of −20 °C until individual sugars and organic acids were analyzed.

### 2.2. Standards and Chemicals

All chemicals, including the standards for sugars and organic acids, were obtained from Sigma–Aldrich (Steinheim, Germany) and Fluka Chemie GmbH (Buchs, Switzerland). All reagents used were of analytical grade of purity.

### 2.3. Sugar Extraction and Analysis

The content of individual sugars (glucose, fructose and sucrose) were analyzed by using a thermal separation technique, i.e., HPLC with a refractive index (RI) detector (Thermo Scientific Finnigan Surveyors ChromQuest 4.0 software). Determination of the content of individual sugars using the HPLC technique was performed according to the Hudina and Stampar [48] method. Ten grams of fresh fruit with bidistilled water totaling a volume of 40 mL was homogenized by using a mixer (Heidolph Promax 202) and left for 30 min at a temperature of 24 °C, as described in research by Begić-Akagić et al. [49]. After extraction, homogenized samples were centrifuged for 7 min at 10,000 rpm at 5 °C with a centrifuge (Thermo Scientific Sorvall ST16R). The resulting extract was filtered through a 0.45 μm cellulose filter (Macherey–Nagel, Düren, Germany), thereby supernatant was obtained, which was transferred directly into vials and used for HPLC analyses of individual sugars and organic acids. Bidistilled water was used to prepare a sugar standard, in order to create a standard curve. The mobile phase (isocratic system) consisted of bidistilled water, while a Hi-Plex Ca column (300 × 7.7 mm, 8 µm; Agilent Technologies, Santa Clara, CA, USA) was used to separate the sugars at a temperature of 65 °C, with a flow rate of 0.6 mL min^−1^ for 30 min. Individual sugars were calculated based on their standards and expressed in g/kg of fresh mass (FM). The glucose and fructose ratio (G/F) was calculated from the values of glucose content and fructose content.

### 2.4. Organic Acids Extraction and Analysis

The content of organic acids (malic, citric, shikimic and fumaric) was analyzed using the HPLC technique, with an ultraviolet (UV) detector with a wavelength of 210 nm, according to the Hudina and Stampar [48] method. Sample extraction was performed according to the method already described for sugars. Bidistilled water was used to prepare an organic acid standard, in order to create a standard curve. For separation of organic acids, the Hi-Plex Ca column (300 × 7.7 mm, 8 µm; Agilent Technologies, Santa Clara, CA, USA) was used at a temperature of 65 °C, with flow rate of 0.6 mL min^−1^ for 30 min, while the mobile phase consisted of 4 mM sulfuric acid. Content of individual acids was calculated based on the acid standards and expressed in g/kg of fresh mass (FM). The content of total organic acids was obtained by summing the data of individual components, and values of total sugar content and total organic acids content were used to calculate the sugar/organic acid ratio (S/A). 

### 2.5. Statistical Analysis

Statistical data processing was performed by using Statgraph 3.14 and SPSS 20 programs. The influence of location and growing year for wild fruit species on content of individual and total sugars and organic acids, as well as S/A and G/F ratios, was analyzed by using two-factor analysis of variance (ANOVA), with the exception of bilberry, where influence of growing year on the parameters noted was tested. The established differences of mean values were tested by Tukey’s test. PCA analysis was used to identify the differentiation factor of wild fruit species from different locations based on analyzed chemical properties (content of individual and total sugars and organic acids, S/A and G/F ratios). For PCA analysis, mean values per year were used. In order to establish the connection of analyzed parameters of wild fruit species quality with environmental factors, Pearson’s correlation coefficient values were calculated by using the SPSS 20 program.

## 3. Results and Discussion

### 3.1. Content of Total and Individual Sugars

The content of total and individual sugars in wild fruit species (rosehip, cornelian cherry, wild strawberry and bilberry) originating from different locations in B&H (Bugojno, Drvar, Foča, Konjic, Romanija, Bjelašnica, Fojnica) is presented in Table 3. A significant influence of location and growing year was established (Table 3), as well as their interaction within the same fruit species (rosehip, cornelian cherry and wild strawberry), i.e., in bilberry, the year of growth on the content of individual and total sugars and G/F ratio. The exceptions were glucose and total sugars in rosehip fruits and fructose in strawberry, where no interaction of experimental factors was established, and G/F ratio in rosehip, where significant differences were not established for growing year. Fructose was predominantly present in analyzed wild fruit species, followed by glucose, while sucrose content was lowest, probably due to high invertase activity in the final ripening stage. This ratio of fructose and glucose in fruit is of particular importance for patients with diabetes, since it maintains the consistency of sugar in blood [50]. Given that the lowest sucrose content was observed in bilberry (1.35 g kg^−1^) in 2015, this fruit should be promoted in a low-sugar diet. Montesano and collaborators [51] reported in all analyzed goji fruits that sucrose content was about ten times lower than each mannose. In analyzed wild fruit, G/F ratio ranged from 0.80 to 0.99, except for wild strawberry samples where this ratio was very low, ranging from 0.35 to 0.51 (Table 3). Such low values of G/F ratio were the result of higher acids content in wild strawberry and not microbial damage. Namely, according to European Fruit Juice Association (AIJN) recommendations [37], strawberry juice with G/F ratio below 0.75 is the indicator of microbial damage. 

In general, content of individual and total sugars and G/F ratio were higher in 2014 than in 2015. The reason for the difference was probably higher average temperature, as well as the amount of precipitation in 2014 in comparison to 2015. On the other hand, by comparing growing regions, it can be clearly seen that the concentrations of mentioned components were higher, e.g., for the Foča region in relation to Konjic concerning rosehip and cornelian cherry and for the Romanija region in relation to Bjelašnica concerning wild strawberry. This could be a consequence of higher average temperatures, higher amounts of precipitation and lower altitudes in comparison to the other analyzed locations (Table 1). As is presented in Table 3, the content of total sugars in analyzed wild fruit was the lowest in bilberry (57.55 g kg^−1^) in 2015, and the highest was in rosehip from the Foča location in 2014 (177.77 g kg^−1^). Within the frame of the same species, the lowest content of total sugars was observed in samples from the Konjic location (138.44 g kg^−1^) in 2015. Depending on fruit species, but also location and growing year, the content of total sugars changed. According to Rosu et al. [52], the total sugar content in rosehip was 132.8 g kg^−1^ in the northeastern region of Romania, and ranged from 133.4 to 171.4 g kg^−1^ for different regions in Iran [53], which is in accordance with the results reported here. However, the values obtained were lower in comparison with results of Barros et al. [54], where sugar content in rosehip was 293.2 g kg^−1^. 

In the research of Tural and Koca [55] and Bijelić et al. [56], the total sugar content in cornelian cherry ranged from 76.80 to 154.00 g kg^−1^ and 134.9 and 252.4 g kg^−1^, respectively, which is in accordance with results of the conducted experiment. However, Drkenda et al. [47] established lower total sugar content in cornelian cherry (62.46–85.20 g kg^−1^). The established differences in total sugar content in cornelian cherry may be a consequence of the amount of precipitation and average temperature during vegetation, as well as type of soil. Therefore, according to Leeuwen et al., [57], climate and soil conditions showed higher impact on the chemical content of fruit in relation to species and cultivar. Values obtained for total sugar content in wild strawberry were lower (75.92–111.56 g kg^−1^) in comparison with Milivojević et al. [40], where total sugar content in wild strawberry was 209.4 g kg^−1^. In addition, lower sugar content was observed in bilberry (57.55 g kg^−1^ in 2015 and 73.75 g kg^−1^ in 2014), in comparison with Milivojević et al. [9] and Milivojević et al. [40]. On the other hand, results obtained for bilberry were in accordance with results obtained by Stajčić et al. [10] and Viljakainen et al. [58], where total sugar content was 78.4 g kg^−1^ and 72.94 g L^−1^ for juice. Lower sugar content in analyzed samples, Romanija wild strawberry (WSR), Bjelašnica wild strawberry (WSB) and Fojnica bilberry (WBF), was followed by higher acid content (Table 4). Therefore, deviations occurring in sugar content compared to the reference data presented appear to be likely caused by differences in location altitudes where the samples were collected, as well as agroecological conditions. 

### 3.2. Content of Total and Individual Organic Acids

Statistical analysis established significant influence of location and growing year as well as their interaction within the same fruit species, and in bilberry the influence of growing year on the content of total and individual organic acids, and S/A ratio. The exception was fumaric acid in wild strawberry, where no influence of analyzed experimental factors was determined, and only growing year had significant influence on shikimic acid within the same fruit species. The interaction of experimental factors did not affect the above-mentioned components in cornelian cherry, citric, fumaric and total acids in rosehip or shikimic and fumaric acids in strawberry fruit (Table 4). The reason for these observations was lower temperatures and lower amount of precipitation in 2015 (Table 2), as well as the higher altitude of growing regions in Konjic (975 m) and Bjelašnica (1430 m) (Table 1). The total acid content in analyzed fruit species ranged from 4.53 g kg^−1^ for wild strawberry from the Romanija location in 2014 and up to 38.17 g kg^−1^ for cornelian cherry in 2015 from the Drvar location.

The statistically significant highest content of total organic acids was observed in cornelian cherry from the Drvar and Konjic locations (36.30 and 35.74 g kg^−1^), followed by cornelian cherry from Bugojno (32.66 g kg^−1^), and the Foča locations having the statistically significant lowest content (28.88 g kg^−1^). These results are in accordance with Begic-Akagic et al. [19], who analyzed influence of location (Goražde, Višegrad and Konjic) on individual organic acid content in cornelian cherry, and total acid content was in the range from 36.6 g kg^−1^ (Drvar) to 48.4 g kg^−1^ (Višegrad). It is presumable that somewhat lower organic acid content present in cornelian cherry from the Foča location was a consequence of climate conditions during the growing season, i.e., higher average temperature (11–11.6 °C) for both the growing years analyzed, in relation to other analyzed locations (Table 1). The lowest total acid content was observed in wild strawberry, which was from 4.53 g kg^−1^ for the Romanija location in 2014 up to 9.22 g kg^−1^ at the Bjelašnica location in growing year 2015. Low organic acid content was also detected in rosehip, which ranged from 5.87 g kg^−1^ for samples from the Foča location in 2014 up to 12.74 g kg^−1^ from the Fojnica location in 2015. Total organic acid content in bilberry was statistically significantly lower in 2014 in comparison to 2015, and it was 9.78 compared to 15.16 g kg^−1^ (Table 4). Demir and Özcan [59] established varying organic acid content in rosehip from different locations in Turkey. Thus, for samples from the Konya (Hadim) location, the acid content expressed as malic acid was 11.7 g kg^−1^, and it was 14.4 g kg^−1^ from the Kastamon location, which is in accordance with the results of the conducted experiment. As per Güneş et al. [23], titratable acidity expressed as citric acid in rosehip was in the range from 11.1 to 36.7 g kg^−1^ and influenced by harvesting time. According to Milivojević et al. [40], the total organic acid content was 0.313 mg (100 g)^−1^ for wild strawberries, which is significantly lower than results of the conducted research (4.53 g kg^−1^ at the Romanija location in 2014 and 9.22 g kg^−1^ at the Bjelašnica location in 2015). Differences observed in organic acid content may be the consequence of location altitude where wild fruit samples were collected, as previously reported by Mikulic Petkovsek et al. [60]. Bilberry fruits from high altitude had more organic acid content compared with bilberry fruit from low altitude. Namely, in the conducted research, wild strawberry was collected from the Romanija location (altitude: 1024 m a.s.l.) and Bjelašnica (1430 m a.s.l.), while according to Milivojević et al. [40], strawberry samples used in the experiment were gathered from locations where the altitude is 350 m a.s.l. According to Mikulic Petkovsek et al. [60], who investigated the content of sugars, organic acids and phenols of 25 genotypes of wild and cultivated fruit in the area of Slovenia, slightly lower concentrations of citric acid and slightly higher concentrations of malic acid (5.7 and 2.71 g kg^−1^) were established in wild blueberries, compared to results of the conducted research (7.07 and 2.56 g kg^−1^ in year of growth 2014). The analyzed berries had a higher total acid content but also lower sugar content (Table 3) compared to reference data, which may be a result of altitude and stage of ripeness, as well as the climate and soil conditions of the environment [23,57,61,62]. The taste and aroma of fruit is directly influenced by S/A ratio. However, the sweet taste of fruit does not always mean higher sugar content, but it is certainly a result of low organic acid content [63]. The highest S/A ratio was recorded in rosehip (30.28) from the Foča location in 2014, and the lowest in cornelian cherry (3.59) in 2015 from the Drvar region. The differences were obvious also amongst locations of rosehip, so the highest S/A ratio was recorded on samples from the Foča location (26.07), followed by Bugojno rosehip (DRB; 22.07), Drvar rosehip (DRD; 17.36) and the significantly lower ratio of the Konjic rosehip sample (DRK; 12.46). Higher S/A ratio was also recorded for wild strawberry in the year 2014 for both locations (15.55 and 24.64). S/A ratio in remaining analyzed samples ranged from 3.59 to 9.01, and it fell under the group of fruit with sweet–sour or sour–sweet taste. The fruit with lower S/A ratio is used for consumption by households [60], and it is mainly processed into a series of highly valuable products such as juices, jams and marmalades [49]. Based on the results obtained, rosehip fruits with higher S/A are recommended for production of “pekmez” [64] and drying, while genotypes of cornelian cherry, wild strawberry and bilberry, with lower S/A, are recommended for the production of juices and gelatin products. 

### 3.3. PCA

According to the results of principal component analysis (PCA), the variability of experiments is explained by three principal components, with 95.9% of total variability, and with components 1 and 2 explaining 84.9% of variability (Figure 1).

Figure 1a shows that the rosehip fruit is positioned in the negative part of component 2. That means that the rosehip fruit from the Bugojno, Drvar and Foča locations is dominantly determined by sucrose, S/A ratio, fructose, total sugars and glucose. The rosehip from the Konjic location is in the positive part of component 3 (Figure 1b), and it is dominantly determined by G/F ratio as well as glucose content. Considering the results presented in Figure 1c, it is evident that rosehip from Foča stands out by its S/A ratio. Regardless of origin, the cornelian cherry is in the negative part of the first component, and shikimic, malic and fumaric acid, glucose, fructose, total acids and total sugars, as well as G/F ratio, dominantly determine the first component. Cornelian cherry samples originating from Drvar and Konjic are located in the positive part of the second component, so in relation to the other two locations, cornelian cherry stands out by its content of citric, malic and fumaric acid, as well as its total acids. 

Wild strawberry and bilberry samples are in the positive part of the first component, and they are dominantly determined by citric acid content, and somewhat less than that by sucrose content and the sugar and acid ratio. However, Figure 1b,c indicate that bilberry in relation to wild strawberry stands out by its glucose content and G/F ratio. On the other hand, in relation to bilberry, strawberries stand out by sucrose content, malic acid and total acids.

### 3.4. Correlation Coefficients

Table 5 presents correlation coefficients between environmental factors (precipitation, average monthly air temperatures and altitude) on the analyzed chemical parameters of cornelian cherry and rosehip. 

By analyzing correlation coefficients of rosehip in relation to temperature, significance was found with respect to content of individual sugars, individual acids (except for fumaric acid), S/A and G/F ratios. Individual sugar content and S/A ratio showed positive correlation with temperature. Conversely, negative correlations between temperature and individual acids and G/F ratio were found. The lowest correlation coefficient −0.74 was in relation to shikimic acid, while the highest, +0.90, was in relation to fructose.

According to altitude, significance was found in relation to all parameters of sugar, malic, citric and total acids, as well as sugar and acid ratio. Malic, citric and total acids showed a positive correlation while each sugar and S/A were negatively correlated with altitude. The lowest significant correlation coefficient, +0.47, was in relation to malic and total acids, while the highest, −0.78, was in relation to fructose and total sugars. Correlation coefficient significance of the analyzed characteristics in relation to annual precipitation was established for only the G/F ratio, and it was −0.53. Analyzing the correlation coefficients of cornelian cherry in relation to temperature, significance was established only in relation to citric acid content, and it was −0.73.

Concerning cornelian cherry, the altitude correlated significantly with glucose, fructose and total sugars, and these correlations were negative, as well as in the case of rosehip. These findings agree with those reported by Ercisli [16], who found that higher total soluble solid (TSS) and total dry weight (TDW) content are desirable fruit characteristics of rosehips, and both characteristics are strongly affected by altitude. 

On the other hand, for cornelian cherry, no correlation coefficient significance was established for the characteristics investigated in relation to annual precipitation. Bijelić et al. [56] established the interdependence of cornelian cherry’s physicochemical parameters and environmental conditions. Significant influence of environmental factors on the synthesis of secondary metabolites in the species *Cornus mas* was established, while synthesis of these components was less influenced by external factors in the case of *Cornus sanguinea* [65]. Considering that wild strawberry and bilberry did not show significant correlation of environmental factors with analyzed parameters of qualities, their results for this study are not presented in Table 5.

## 4. Conclusions

Analyzed individual sugars and organic acids were detected in all wild fruit species. An influence of location and growing year was established within the same fruit species (cornelian cherry, rosehip and wild strawberry), and growing year for bilberry, on individual and total sugars and organic acids, which are the most important factors in the characterization of wild fruit from a nutritive perspective, as well as for processing the fruit into valuable products and proving their authenticity. 

Based on the S/A ratio results, rosehip fruits are recommended for the production of “pekmez” and drying, while genotypes of cornelian cherry, wild strawberry and bilberry, with lower S/A, are suitable for the production of juices and gelatin products. A significant influence of environmental factors on the synthesis of analyzed primary metabolites in cornelian cherry and rosehip species was established, while this influence was not established in the case of wild strawberry and bilberry species. 

The results obtained on wild fruit species expand knowledge about the nutritionally valuable components that contribute to health protection and that the fruit may be widely used in the food industry as functional food as well as in medicine. According to these findings, it is necessary for future research to focus on the detection and quantification of polyphenol components in this valuable plant material, which can prevent the occurrence of disease and ensure a better quality of life. 

## Figures and Tables

**Figure 1 foods-09-00462-f001:**
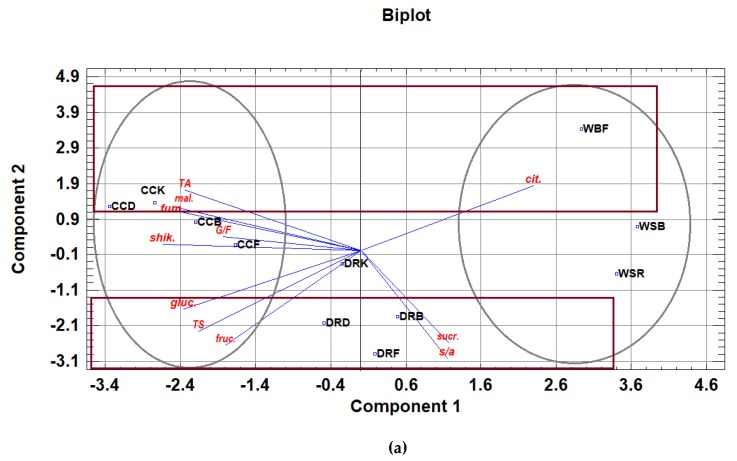
Principal component analysis (PCA) plots presenting graphical representation of the position of the analyzed wild fruit species from different growing regions (DRB, DRD, DRF, DRK—rosehip from Bugojno, Drvar, Foča and Konjic locations; CCB, CCD, CCF, CCK—cornelian cherry from Bugojno, Drvar, Foča and Konjic locations; WSR, WSB—wild strawberry from Romanija and Bjelašnica locations; WBF—wild bilberry from the Fojnica location) in relation to sugars (gluc—glucose; fruc—fructose; sucr—sucrose; TS—total sugars), organic acids (TA—total acids; mal—malic; cit—citric; shik—shikimic; fum—fumaric acid), glucose/fructose ratio (G/F) and sugar/acid ratio (s/a). (**a**) component 1 vs. component 2; (**b**) component 1 vs. component 3; (**c**) component 2 vs. component 3.

**Table 1 foods-09-00462-t001:** Monthly meteorological data for the growing regions in Bosnia and Herzegovina that were analyzed.

Month/Growing Season		Location/Meteorological Date													
		Bugojno *		Drvar *		Foča **		Konjic *		Romanija **		Bjelašnica *		Fojnica *	
		Tm	Ppt	Tm	Ppt	Tm	Ppt	Tm	Ppt	Tm	Ppt	Tm	Ppt	Tm	Ppt
January	2014	4.6	70.4	4.7	125.7	4.1	52.1	3.1	162.7	2.0	37.5	−2.7	91.0	5.0	55.3
	2015	0.1	90.1	1.6	121.9	0.6	161.5	−1.0	222.9	−2.5	89.9	−5.7	173.4	0.9	112.6
February	2014	7.2	41.1	5.8	195.1	6.9	20.5	4.7	74.9	4.4	10.8	−2.7	78.7	7.8	19.9
	2015	1.3	43.9	1.2	59.9	2.2	52.6	−0.7	109.9	−1.3	56.1	−6.9	131.0	1.7	56.6
March	2014	7.8	34.7	7.3	59.0	8.2	74.6	5.4	109.5	4.8	63.6	−2.5	109.0	8.1	67.3
	2015	5.3	55.1	5.3	46.4	5.4	91.2	2.3	126.7	2.1	74.7	−5.5	143.8	5.3	80.4
April	2014	10.5	153.6	10.8	178.9	11.4	156.3	7.6	184.9	7.6	153.9	−0.1	210.7	10.2	148.5
	2015	9.1	47.7	9.2	26.6	9.6	54.6	6.2	83.8	6.2	53.2	−2.8	73.3	9.2	43.6
May	2014	13.4	125.7	13.1	195.9	14.1	118.4	9.9	188.9	10.7	260.7	3.1	185.8	13.5	186.2
	2015	15.6	60.0	15.3	111.1	16.5	51.9	12.9	41.2	13.4	75.4	5.6	38.2	16.1	52.9
June	2014	17.6	96.0	17.9	116.3	18.4	123.9	14.5	168.8	15.2	170.0	7.6	170.1	17.5	125.1
	2015	18.0	72.6	18.0	55.9	18.2	121.3	14.9	94.9	15.3	113.3	7.7	118.6	17.8	91.0
July	2014	18.7	161.5	19.0	89.5	19.5	134.1	16.1	112.3	17.0	76.2	9.5	198.5	19.5	73.3
	2015	22.7	67.6	22.7	25.2	22.7	23.2	19.7	21.2	19.8	25.2	13.9	25.7	23.2	9.4
August	2014	19.2	67.5	19.4	81,2	19.8	93.8	16.5	94.9	16.8	112.6	10.5	70.9	19.7	78.2
	2015	21.3	51.3	21.2	33.7	21.9	30.1	18.7	63.6	18.9	85.7	12.8	68.4	21.8	57.4
September	2014	14.7	173.8	14.5	240.3	15.7	142.8	12.3	270.8	12.2	147.0	5.9	215.4	14.9	139.5
	2015	16.1	91.2	16.0	107.2	17.7	50.7	14.4	102.2	14.5	84.3	7.8	77.8	17.6	60.2
October	2014	11.8	49.7	11.4	69.8	11.5	58.5	9.4	71.5	9.1	42.9	4.0	92.1	12.0	59.0
	2015	10.9	166.6	11.0	185.6	11.6	145.8	8.8	266.2	8.4	106.9	3.0	315.4	11.1	124.6
November	2014	8.3	55.2	8.8	170.6	7.5	42.1	7.0	131.8	6.0	56.6	1.3	139.4	8.9	47.0
	2015	4.5	62.5	5.0	45.2	5.2	83.0	5.4	118.0	3.4	71.2	2.5	178.4	6.0	75.1
December	2014	2.2	85.2	2.7	133.9	2.6	116.0	0.8	137.2	−0.5	88.0	−4,4	211.1	2.7	83.4
	2015	−1.3	0.2	0.8	0.2	0.6	0.6	1.2	2.2	−1.8	0.6	−0.2	1.6	−0.5	2.5
**Average year Tm/ ∑Ppt**	**2014**	11.3	1114.4	11.3	1656.2	11.6	1133,1	9.0	1708.2	8.8	1219,80	2.5	1772.7	11.6	1082.7
	**2015**	10.3	808.8	10.6	819.9	11.0	866,50	8.6	1252.8	8.0	836,50	2.7	1345.6	10.8	766.3

* data from Federal Hydrometeorological Service, Sarajevo; ** Republic Hydrometeorological Service Banja Luka; Tm—mean monthly air temperature (°C); Ppt—quantity of rainfall (mm).

**Table 2 foods-09-00462-t002:** Sample codes of wild fruit species.

Genotype	Growing Region	Sample Code	Harvesting Time 2014/2015	Color
**Rosehip** **(*Rosa canina* L.)**	Bugojno (altitude = 942 m a.s.l.*, 43°19′23.74′’N /18°8′16.33′’E carbonated brown soil)	DRB	12.10./25.10.	bright red
	Drvar (altitude = 700 m a.s.l., 44°22′26′’N /16°22′50.9′’E carbonated red soil)	DRD	3.10./5.10.	
	Foča (altitude =511 m a.s.l., 43°35′3′’N /18°47′33′’E acidic brown soil)	DRF	30.9./3.10.	
	Konjic (altitude = 975 m a.s.l., 43,83°N /17,85°E district brown soil)	DRK	9.10./28.10.	
**Cornelian cherry** **(*Cornus mas* L.)**	Bugojno (altitude = 942 m a.s.l., 43°19′23.74′’N /18°8′16.33′’E carbonated brown soil)	CCB	25.9./28.9.	dark red
	Drvar (altitude = 700 m a.s.l., 44°22′26′’N /16°22′50.9′’E carbonated red soil)	CCD	23.9./27.9.	
	Foča (altitude = 511 m a.s.l., 43°35′3′’N /18°47′33′’E acidic brown soil)	CCF	15.9./19.9.	
	Konjic (altitude = 750 m a.s.l., 43.5807°N /18.0166°E carbonated brown soil)	CCK	20.9./26.9.	
**Wild strawberry (*Fragaria vesca* L.)**	Romanija (altitude = 1024 m a.s.l., 43°52′33.2′’N /18°40′10.92′’E rendzina)	WSR	20.6./29.6.	red
	Bjelašnica (altitude = 1430 m a.s.l., 43,67°N /18,23°E rendzina)	WSB	5.7./8.7.	
**Bilberry** **(*Vaccinium myrtillus* L.)**	Fojnica (altitude = 1670 m a.s.l., 43°57′26′’N /17°45′17′’E brown acidic soil)	WBF	16.8./24.7.	dark purple

* Above sea level.

**Table 3 foods-09-00462-t003:** Average content of individual sugars, total sugars (TS) and glucose/fructose ratio (G/F) ± SE in analyzed wild fruit species (g kg^−1^ FM (fresh mass)) *.

Species/Growing Region	Growing Season	Glucose		Fructose		Sucrose		TS		G/F	
		x̄	SE	x̄	SE	x̄	SE	x̄	SE	x̄	SE
DRB	2014	70.78	1.777x	84.66	2.317x	4.66	0.234x	160.11	2.166x	0.84	0.027
	2015	65.51	0.658y	78.53	0.396y	3.98	0.019y	148.02	0.970y	0.83	0.007
*Mean by region*		*68.15*	*3.126b*	*81.60*	*3.471b*	*4.32*	*0.399b*	*154.06*	*6.787b*	*0.84*	*0.018c*
DRD	2014	80.38	1.455	91.45	1.091x	5.47	0.456x	177.31	0.276x	0.88	0.026
	2015	74.01	0.735	86.96	1.123y	4.63	0.027y	165.60	0.236y	0.85	0.002
*Mean by region*		*77.19*	*4.610a*	*89.20*	*2.557a*	*5.05*	*0.546a*	*171.45*	*6.417a*	*0.87*	*0.023b*
DRF	2014	78.96	0.619	92.68	1.287x	6.12	0.294	177.77	2.146x	0.85	0.007
	2015	73.28	0.773	86.62	0.360y	4.30	0.012	164.20	0.910y	0.85	0.002
*Mean by region*		*76.12*	*2.173a*	*89.65*	*3.428a*	*5.21*	*1.013a*	*170.99*	*7.574a*	*0.85*	*0.008bc*
DRK	2014	70.89	0.959	78.84	0.793x	4.18	0.151	153.92	1.213x	0.90	0.017
	2015	66.07	0.059	69.33	1.481y	3.05	0.047	138.44	0.484y	0.95	0.006
*Mean by region*		*68.48*	*2.712b*	*74.09*	*5.244c*	*3.61*	*0.629c*	*146.18*	*8.516c*	*0.93*	*0.032a*
CCB	2014	74.61	1.437x	81.44	0.800x	4.03	0.181x	160.08	0.726x	0.92	0.026x
	2015	67.78	0.420y	71.21	0.251y	2.62	0.060y	141.61	0.443y	0.95	0.007y
*Mean by region*		*71.19*	*3.856a*	*76.33*	*5.629c*	*3.33*	*0.786b*	*150.85*	*10.134b*	*0.93*	*0.026a*
CCD	2014	75.90	0.784x	82.11	0.411x	3.79	0.256x	161.79	0.354x	0.92	0.013x
	2015	62.49	0.873y	71.64	0.307y	2.89	0.088y	137.03	1.218y	0.87	0.009y
*Mean by region*		*69.19*	*7.380b*	*76.87*	*5.74bc*	*3.34*	*0.521b*	*149.41*	*13.589b*	*0.90*	*0.03b*
CCF	2014	71.90	1.607x	84.18	1.510x	4.84	0.332x	160.93.	2.675x	0.85	0.013x
	2015	66.04	0.141y	78.80	0.319y	3.17	0.007y	148.01	0.432y	0.84	0.003y
*Mean by region*		*68.97*	*3.370b*	*81.49*	*3.103a*	*4.00*	*0.938a*	*154.47*	*7.276a*	*0.85*	*0.012c*
CCK	2014	70.32	1.039x	80.62	1.276x	3.42	0.106x	154.36	2.322x	0.87	0.002x
	2015	60.04	0.053y	74.86	0.084y	2.72	0.105y	137.61	0.186y	0.80	0.001y
*Mean by region*		*65.18*	*5.667c*	*77.74*	*3.26b*	*3.07*	*0.398b*	*145.99*	*9.292c*	*0.84*	*0.034c*
WSB	2014	24.54	1.170x	62.83	1.637x	5.18	0.172x	92.55	0.468x	0.39	0.029x
	2015	18.50	0.383y	52.71	1.089y	4.70	0.007y	75.92	1.363y	0.35	0.006y
*Mean by region*		*21.52*	*3.398b*	*57.77*	*5.682b*	*4.94*	*0.280b*	*84.23*	*9.156b*	*0.37*	*0.029b*
WSR	2014	35.43	1.030x	69.30	0.736x	6.84	0.134x	111.56	1.047x	0.51	0.017x
	2015	22.60	0.300y	58.34	0.458y	5.94	0.186y	86.87	0.276y	0.39	0.008y
*Mean by region*		*29.01*	*7.060a*	*63.82*	*6.025a*	*6.00*	*1.332a*	*99.22*	*13.539a*	*0.45*	*0.069a*
WBF	2014	34.40	0.587x	36.49	1.126x	2.86	0.130x	73.75	1.794x	0.94	0.016
	2015	27.98	0.856y	28.22	1.048y	1.35	0.120y	57.55	1.787y	0.99	0.031
*Mean by year*		*31.19*	*3.573*	*32.35*	*4.631*	*2.11*	*0.838*	*65.65*	*9.014*	*0.97*	*0.035*

* Average values ± standard error (SE) in column marked with different letters (a–c) represent statistically significant differences between locations within the same species; different letters (x–y) represent statistically significant differences between growing year of wild fruit; Tukey’s test, *p* < 0.05.

**Table 4 foods-09-00462-t004:** Average content of individual organic acids, total acids (TA) and sugars/acids ratio (S/A) in analyzed wild fruit species (g kg^−1^ FM) *.

Species/Growing Region	Growing Season	Malic Acid		Citric Acid		Shikimic Acid		Fumaric Acid		TA		S/A	
		x̄	SE	x̄	SE	x̄	SE	x̄	SE	x̄	SE	x̄	SE
DRB	2014	4.09	0.124x	1.57	0.114x	0.33	0.042x	0.07	0.006x	6.07	0.253x	26.42	1.161x
	2015	5.87	0.057y	1.96	0.910y	0.43	0.043y	0.09	0.007y	8.35	0.015y	17.73	0.084y
*Mean by region*		*4.98*	*1.230c*	*1.76*	*1.023c*	*0.38*	*0.066d*	*0.08*	*0.013c*	*7.21*	*1.260c*	*22.07*	*4.816b*
DRD	2014	6.43	0.223x	2.06	0.161x	0.52	0.042x	0.25	0.025x	9.26	0.202x	19.16	0.424x
	2015	7.27	0.183y	2.27	0.987y	0.78	0.009y	0.34	0.008y	10.66	0.453y	15.55	0.662y
*Mean by region*		*6.85*	*0.493b*	*2.16*	*0.560b*	*0.65*	*0.146b*	*0.29*	*0.051a*	*9.96*	*0.830b*	*17.36*	*2.038c*
DRF	2014	3.87	0.098x	1.48	0.024x	0.44	0.028x	0.07	0.003x	5.87	0.079x	30.28	0.113x
	2015	5.11	0.075y	1.80	0.005y	0.53	0.004y	0.08	0.001y	7.51	0.080y	21.86	0.177y
*Mean by region*		*4.49*	*1.679d*	*1.64*	*0.210c*	*0.49*	*0.049c*	*0.08*	*0.010c*	*6.69*	*0.902d*	*26.07*	*4.616a*
DRK	2014	7.35	0.184x	2.78	0.117x	0.71	0.034x	0.12	0.024x	10.96	0.318x	14.06	0.462x
	2015	8.55	1.567y	3.18	0.155y	0.84	0.031y	0.18	0.069y	12.74	0.369y	10.87	0.346y
*Mean by region*		*7.95*	*3.093a*	*2.98*	*0.340a*	*0.77*	*0.078a*	*0.15*	*0.057b*	*11.85*	*1.027a*	*12.46*	*1.78d*
CCB	2014	28.61	0.863x	1.30	0.044x	0.57	0.121x	0.50	0.060x	30.97	0.832y	5.17	0.126x
	2015	31.43	0.582y	1.49	0.098y	0.83	0.013y	0.59	0.02y	34.34	0.532x	4.12	0.075y
*Mean by region*		*30.02*	*1.681b*	*1.40*	*0.124a*	*0.70*	*0.158b*	*0.55*	*0.066b*	*32.66*	*1.945b*	*4.65*	*0.580b*
CCD	2014	30.81	1.929x	1.43	0.039x	1.18	0.242x	1.01	0.079x	34.43	1.687y	4.71	0.218x
	2015	34.06	1.102y	1.54	0.067y	1.45	0.386y	1.12	0.036y	38.17	0.804x	3.59	0.066y
*Mean by region*		*32.44*	*2.269a*	*1.48*	*0.080a*	*1.31*	*0.322a*	*1.07*	*0.082a*	*36.30*	*2.365a*	*4.15*	*0.628c*
CCF	2014	24.98	0.884x	1.13	0.112x	0.51	0.028x	0.36	0.015x	26.99	0.794y	5.96	0.098x
	2015	28.31	0.457y	1.30	0.145y	0.73	0.009y	0.42	0.007y	30.76	0.400x	4.81	0.051y
*Mean by region*		*26.65*	*1.931c*	*1.22*	*0.135b*	*0.62*	*0.118b*	*0.39*	*0.037b*	*28.88*	*2.140c*	*5.39*	*0.635a*
CCK	2014	31.13	0.938x	1.43	0.171x	0.73	0.041x	1.02	0.109x	34.31	0.662y	4.50	0.086x
	2015	33.39	0.564y	1.63	0.088y	0.96	0.013y	1.19	0.302y	37.17	0.935x	3.70	0.092y
*Mean by region*		*32.26*	*2.034a*	*1.53*	*0.163a*	*0.85*	*0.128b*	*1.10*	*0.223a*	*35.74*	*1.725a*	*4.10*	*0.443c*
WSB	2014	1.19	0.011y	4.70	0.407y	0.08	0.010y	0.001	0.0001	5.97	0.396y	15.55	1.124x
	2015	2.13	0.023x	6.97	0.115x	0.12	0.014x	0.002	0.0001	9.22	0.094y	8.23	0.172y
*Mean by region*		*1.66*	*0.518b*	*5.83*	*1.271a*	*0.10*	*0.022*	*0.002*	*0.0002*	*7.60*	*1.799a*	*11.89*	*4.07b*
WSR	2014	1.23	0.103y	3.21	0.102y	0.09	0.008y	0.002	0.0005	4.53	0.178y	24.64	1.20x
	2015	2.68	0.042x	6.07	0.042x	0.11	0.002x	0.002	0.0006	8.86	0.075x	9.91	0.073y
*Mean by region*		*1.95*	*1.123a*	*4.64*	*1.57b*	*0.10*	*0.011*	*0.002*	*0.0005*	*6.70*	*2.37b*	*17.22*	*5.159a*
WBF	2014	2.56	0.250y	7.07	0.237y	0.14	0.013y	0.0010	0.0002y	9.78	0.471y	7.55	0.317x
	2015	4.35	0.145x	10.63	0.262x	0.18	0.007x	0.0014	0.0000x	15.16	0.254x	3.80	0.183y
*Mean by year*		*3.45*	*1.340*	*8.85*	*1.96*	*0.16*	*0.022*	*0.001*	*0.0003*	*12.47*	*2.968*	*5.67*	*2.068*

Average values ± standard error (SE) in column marked with different letters (a–c) represent statistically significant differences between locations within the same species; different letters (x–y) represent statistically significant differences between growing year of wild fruit; Tukey’s test, *p* < 0.05.

**Table 5 foods-09-00462-t005:** Correlation coefficients between rainfall, temperature, altitude and the content of sugars and organic acids.

		Sucr	Gluc	Fruc	TS	G/F	Mal	Shik	Cit	Fum	TA	S/A
Tm	DR	**0.80 ****	**0.68 ****	**0.90 ****	**0.83 ****	**−0.75 ****	**−0.83 ****	**−0.74 ****	**−0.92 ****	−0.11	**−0.85 ****	**0.85 ****
	CC	0.62	0.60	0.42	0.56	0.42	−0.64	−0.14	**−0.73(*)**	−0.66	−0.62	0.65
Alt	DR	**−0.65 ****	**−0.74 ****	**−0.78 ****	**−0.78 ****	0.38	**0.47 ***	0.20	**0.55 ****	−0.05	**0.47 ***	**−0.56 ****
	CC	−0.66	**−0.75 ***	**−0.78 ***	**−0.78 ***	0.40	0.47	0.20	0.57	−0.05	0.47	−0.56
Ppt	DR	0.13	0.30	−0.03	0.11	**−0.53 ****	0.29	0.17	0.38	0.07	0.31	−0.20
	CC	0.14	0.30	−0.03	0.11	0.56	0.29	0.17	0.39	0.07	0.31	−0.20

DR—dog rose; CC—cornelian cherry; Gluc—glucose; Fruc—fructose; Sucr—sucrose; TS—total sugar; TA—total acid; Mal—malic; Cit—citric; Shik—shikimic; Fum—fumaric acid; G/F—glucose/fructose ratio; S/A—sugar/acid ratio; Tm—mean monthly air temperature (°C); Ppt—quantity of rainfall (mm); Alt—altitude (m); *—significance at *p* ≤ 0.05; **—significance at *p* ≤ 0.01.

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
