# Peer review of "Geographic Variability of Sugars and Organic Acids in Selected Wild Fruit Species"

_foods, 2020, doi:10.3390/foods9040462_

Round 1

Reviewer 1 Report

Dear Authors,

Abstract:

The authors should explain the acronyms: HPLC, PCA.

Line 24: please, change “determined” with another word.

I suggest to the authors to shorten the abstract in order to be it more concise. For example, no results about PCA are reported, but general considerations, useful for the conclusions or for the final part of R&D.

Keywords: please delete “individual” and “possibility of”. ….sugars, organic acids,…

Introduction:

lines 77-81: I suggest to change x), xx), xxx)…with i); ii); iii); iv)….

M&M

Table2: please explain a.s.l. (above sea level) as footnote

Why the genotype Wild Strawberry are two? And Bilberry only one?

Can you add to Table 2 where come from the samples, so can you add the geographical coordinates for Bugojno, Drvar…and the others? The altitude that i find on Internet are different. Why?

R&D

In order to have a better comparison with literature data, I strongly suggest to the authors to add the following reference regarding a simple HPLC-ELSD procedure to analyze sugars in berry: https://doi.org/10.1155/2016/6271808

Lines 242, 243: please delete the point after m. For example: …. (1024 m a.s.l.)…

Line 269: Figure 1 or Figure 1a?

I suggest to the authors to ameliorate the Figures of PCA plots, in particular when the names of the vectors are very near.

Conclusions:

I suggest to the authors to shorten the conclusions in order to be it more concise. In my opinion, the authors should highlight that the present research in the future will regard other bioactives, typical of this fruits, for example phenols, only shortly mentioned in the Introduction.

Author Response

Dear Reviewer 1,

Please see in the attachment.

Reviewer 2 Report

Dear Authors:

Your work is really interesting and provide information about wild berries to be used as ingredients in many products, as you mention.

It was a hard work to collect fruits during different periods, prepare samples, analyze them and finally process data.

Introduction section, some info about why do you choose these four wild berries would be useful.

Material and methods section need to be revised and improved as I do some suggestions.

Results and discussion section should also be revised and take into account my suggestions.

Finally conclusion section will answer those four aims.

Although your hard work, you need to ameliorate it with some of my comments made in the attached file.

Hope these comments and suggestions will serve to facilitate the better understanding of your manuscript.

Thanks.

Author Response

Dear Reviewer,

Reviewer 3 Report

The article describes the variation in the secondary metabolite content (sugars and organic acids) in some of the wild fruits and their relation to geographical conditions.

The article is very exhaustive and does not show any focus. It has nutrition, biodiversity, phytogeography, etc. So, the content is diluted. Though the title says Influence on the processing, the methods does not describe about any processing of the specimens. Preparation of samples for biochemical estimations cannot be considered as processing.

No logic presented for selecting these fruits. Are they the only available wild fruits or they have great market value or etc etc...

It is very difficult to draw conclusions based on 2 years data in these kind of studies. As you may require 4-5 years data to draw a conclusion. So, I do not see any novel observation coming from this study.

It is also difficult to conclude on the nutritional value of the fruit by just sugar and organic acid content. There are other important phytochemicals like polyphenols etc.

Generally, the extraction and characterization methods are different for sugars and organic acids. So, methods should have had separate sections for extraction and characterization of these substances.

The authors can consider presenting individual fruits, with more sampling for multiple years and more phytochemicals. 

The current article does not have any new/significant information for the readers.

Author Response

Dear Reviewer,

Round 2

Reviewer 3 Report

I do not see a point to point clarification on the review queries with the revised version, so I am not able to see what changes have been done by the authors from the previous version.

Though the article starts with four different wild fruits, but much the data presented is with two fruits (Rosa canina and cornelian cherry), so the authors can decide on whether to continue presenting data from the lesser studied varieties. I do not see those varieties contributing to the conclusions. This will increase the article focus.

Its better to present data on both the fruits as separate tables or figures. Because they have no inter-relation and they are two different varieties. Hope all statistics done is independent for each variety.

Author Response

Dear Reviewer,
